# Changes in Wolf Occupancy and Feeding Habits in the Northern Apennines: Results of Long-Term Predator–Prey Monitoring

**DOI:** 10.3390/ani14050735

**Published:** 2024-02-27

**Authors:** Elisa Torretta, Anna Brangi, Alberto Meriggi

**Affiliations:** Department of Earth and Environmental Sciences, University of Pavia, Via Ferrata 1, 27100 Pavia, Italy; anna.brangi@gmail.com (A.B.); alberto.meriggi@unipv.it (A.M.)

**Keywords:** dynamic occupancy model, scat analysis, *Canis lupus*, range expansion, *Capreolus capreolus*

## Abstract

**Simple Summary:**

The ongoing range expansion of the wolf (*Canis lupus*) in Italy is currently leading the species closer to highly anthropized landscapes. Long-term research projects carried out in focal areas along the Apennines provide the opportunity to finely monitor the dynamics of this expansion and to predict the new scenarios of wolf–human coexistence. We carried out a monitoring project (2007–2022) based on the collection of indirect signs of presence along routes in an area of the Northern Apennines where both the wolf’s range and its feeding habits greatly changed. Wolves first settled in the mountains of our study area and, once it had been saturated, they began to occupy the hill zones. These recently occupied zones are markedly different from the undisturbed mountains: woodlands are restricted to small patches, and agricultural lands are predominant. Nevertheless, wolves rapidly colonized the hills, driven by roe deer (*Capreolus capreolus*) abundance. This widespread species became very abundant during the last few decades and supported the wolf range expansion; indeed, its consumption greatly increased until it became the staple of the wolf diet. Our results suggest that wolf expansion will follow that of the roe deer across the Po Plain with the settlement of wolf packs in roe deer high-density areas.

**Abstract:**

The comprehension of the factors that have influenced the recent changes in wolf (*Canis lupus*) range and diet that have occurred in our study area, characterized by a highly heterogeneous landscape, can shed light on their current process of expansion toward the plain. Wolf presence was monitored using a standardized protocol from 2007 to 2022 by carrying out eight monitoring sessions organized in seasonal surveys, during which, we collected wolf presence data. To model wolf range dynamics, we used dynamic occupancy models considering land cover types and wild ungulate abundances as covariates. Moreover, we studied the wolf diet through scat analysis, identifying the consumed items from undigested remains. Wolf occupancy in the study area progressed from mountains to lower hills gradually; the observed range dynamics were driven by prey abundance and human presence: in particular, the probability of colonization increased with roe deer (*Capreolus capreolus*) abundance, whereas the probability of extinction increased with urban areas. The wolf diet showed a gradual shift from the prevalent consumption of wild boar (2007–2008 and 2011–2012) to the prevalent consumption of roe deer (continuously increasing from 2015 onward). Our results might be related to a specific adaptation of the predator to the local ecology of the most consumed species: the roe deer.

## 1. Introduction

Coexistence with large carnivores has always generated considerable conflicts, and their conservation and management have been highly controversial and often politicized [1]. In European-human-dominated landscapes, where top predators have been on the rise for many decades [2,3,4], social conflicts are becoming more frequent and more complex (e.g., depredation of pets [5]), following the general expansion of the species toward highly urbanized areas.

One of the largest wolf (*Canis lupus*) populations persisting in Europe is located in the Italian Peninsula, which is estimated to be ca. 3300 wolves (range: 2945–3608) distributed over ca. 150,000 km^2^ [6].

The Apennines, the mountain system extending for ca. 1200 km along the Italian Peninsula, host the largest portion of this population, corresponding to ca. 2400 wolves (range: 2020–2645) continuously distributed across the entire mountain system, from the Aspromonte in the south (Calabria region) to the Ligurian and Piedmont Apennines in the north [6]. The species is widespread along mountainous valleys and ridges, i.e., the most undisturbed yet saturated areas [7], and its expansion is currently ongoing in flat areas, such as the urbanized and agricultural Po Plain [8,9], and in coastal areas, such as the Maremma in southern Tuscany [10]. Thus, both the size and range of the Italian wolf population have recently increased, leading to a reassessment of the species’ status in Italy (from Vulnerable in 2013 to Nearly Threatened in 2022 [11]).

Apart from a recent national survey [6], many short-term and often small-scale research projects have been conducted all over Italy using heterogeneous methods [12]. Long-term research projects are challenging and barely implementable, as they usually require prolonged sampling efforts, consistency in methods throughout the sampling period, adequate data analyses, and durable funding [13]. Such projects are particularly scarce in the human-dominated landscapes of southern Europe [14], but the dynamics of wolf recolonization along the Italian Apennines, started in small and isolated areas of the Central and Southern Apennines and was completed over a couple of decades [15], offered interesting case studies in this regard.

After a dramatic human-caused decline, which reduced the population to less than a hundred individuals [16,17], wolves reached the most northern sector of the Apennines at the border between Liguria, Piedmont, Lombardy, Emilia-Romagna, and Tuscany in the 1980s [18]. The initial phase of the recolonization process was slowed by two main factors: prey availability and illegal killing. In particular, among wild ungulates, the wild boar (*Sus scrofa*) was the only widespread species, while deer were generally scattered or rare [18,19]. The presence of herds on mountain pastures during the free grazing period, generally from May to October, and the lack of protective measures and irregular surveillance, made domestic ungulates a seasonally important food resource, which facilitated the settlement of individuals [20]. At the same time, the depredation of livestock triggered reprisal episodes. The presence of the first packs in the Lombard Apennines was detected between 1985 and 1990, but their presence was irregular across the sampling sessions because of recurrent poaching events [18,19].

Thanks to a long-term monitoring project carried out in an area of the Northern Apennines located in the Lombardy region, we are able to retrace the dynamics that have led to the current situation. The Lombard Apennines represent a focal area for the study of the wolf in Italy because they have been monitored since the late 1980s [9,18,19,20,21,22,23,24,25] and because of their strategic location; lying along the Apennine mountain chain and facing the Po Plain near its most functional ecological corridor (e.g., the Ticino River [8]), they represent a sort of junction for the expanding wolf population. Thus, the comprehension of factors that have influenced the recent changes in wolf range and diet may clarify the ongoing process of expansion. Indeed, although there are notable differences in landscape composition and prey availability between the Apennines and the Po Plain [8], the main drivers of wolf expansion, such as human presence, woodlands, and prey species, just to mention a few, could play a comparable role in both areas.

The aims of this study were to analyze (a) range dynamics and (b) changes in wolf feeding habits that occurred during the period 2007–2022. Regarding our first aim, we predicted that (i) the range expansion would follow a specific gradient from less human-inhabited zones to more human-dominated ones and that (ii) different factors would drive the different stages of wolf range expansion, with the availability of natural habitats driving the early stages within less human-inhabited zones and the magnitude of human pressure driving the recent stages within more human-dominated zones. As for our second aim, we predicted that, following the general trend observed in Europe [9], (iii) the wolf diet would increasingly be dominated by wild ungulate consumption, resulting in a narrowing diet breadth. We expected a reduction in livestock consumption both in less human-inhabited zones due to the availability of wild prey and livestock protection measures and in more human-dominated zones where livestock is typically housed. Additionally, (iv) among the available ungulate species, we predicted that the roe deer (*Capreolus capreolus*) would constitute the staple of the wolf diet because of the species’ range expansion and its increased profitability for the predator.

## 2. Materials and Methods

### 2.1. Study Area

The Lombard Apennines are a restricted portion of the Northern Apennines, localized in the southwestern part of the Lombardy region, which lies to the south of the Po River. The study area spreads over 850 km^2^ and ranges from 200 m to 1724 m a.s.l. (M. Lesima). The climate is sub-Mediterranean with an average yearly temperature of 9 °C (January: av. temp. = −1.1 °C, July: av. temp. = 18.1 °C). Mean annual precipitation ranges from 950 mm at lower elevations to 1300 mm at higher elevations, mostly concentrated in two rainy seasons (April–May and November). The landscape is very heterogeneous; considering the topography, the study area can be subdivided into three elevational zones well characterized by different land cover classes. The northern portion of the study area, localized just behind the Po Plain, is the lower-hill zone, which ranges from 200 m to 500 m a.s.l. and is characterized by cultivated lands. Vineyards and annual crops are predominant (37.6% and 26.2% of the surface, respectively) and urbanized area covers 10.3% of the surface. Woodlands are restricted to small patches along streams (14.7%), and uncultivated lands cover 5.5% of the surface. It follows the upper-hill zone, which ranges from 500 to 800 m a.s.l. and where annual crops are predominant (24.8%, mainly fodder crops such as cereals and leguminous plants), but vineyards are still present (11.5%), and uncultivated lands reach 8.8%. Broad-leaved woodlands cover 40.0% of the surface. The most representative species are *Quercus* spp. and *Castanea sativa*. Urban areas represent 4.2% of the surface. The southern portion of the study area is the mountainous zone (>800 m a.s.l.) mainly covered by natural vegetation (78.7%), where broad-leaved woodlands (54.7%; main species: *Fagus sylvatica*) are predominant on conifer reforestations (1.5%; *Pinus* spp. and *Picea abies*) or mixed woodlands (1.3%). Uncultivated lands reach 10.2%, and pastures–meadows reach 7.4%. Cultivated lands cover 14.3% of the surface (mainly annual crops), and urban areas are restricted to 2.8% of the surface (Figure 1). The average human population density of the whole study area is 27.43 inhabitants per km^2^. 

A rich wild ungulate community is present in the study area, composed of four species: the wild boar (*Sus scrofa*), introduced for hunting; the roe deer (*Capreolus capreolus*), which has gradually increased during the last few decades and is currently widespread and very abundant; the fallow deer (*Dama dama*), which is scattered within the study area with high-density spots; and lastly, the red deer (*Cervus elaphus*), which is present in the southern zone of the study area with an increasing population. Livestock represents a secondary economic activity in the area, even though, in a few decades, it shifted from the extensive farming of free-grazing cows to the intensive rearing of stall-fed sheep and goats (in 2010: 4100 cows and 600 sheep and goats; in 2017: 2700 cows and 1100 sheep and goats); extensive farming persists especially in the mountainous zone and is characterized by free-grazing livestock on pastures, generally from May to October [24].

Within the study area, the first unquestionable evidence of wolf recolonization was an illegally killed individual found in 1987; the results obtained by the first research project (1987–1992) reported the presence of two breeding pairs with pups along the southernmost mountain ridge [18].

### 2.2. Data Collection

Wolf presence was monitored using a standardized protocol from 2007 to 2022 by carrying out eight monitoring sessions; each session of data collection included at least two seasonal surveys, but generally, it consisted of four seasonal surveys covering an entire year (winter: December–February; spring: March–May; summer: June–August; autumn: September–November). We were not able to carry out the monitoring continuously during the study period; thus, some years (e.g., 2013, 2014, 2016, 2017) are lacking.

During each seasonal survey, we searched for indirect signs of wolf presence along routes. We adopted a Tessellation Stratified Sampling design [26], subdividing the study area into sample squares of 25 km^2^ (5 × 5 km) and randomly selecting at least one route among the existing footpaths and dirt roads within each square. Thus, selected routes were representative of the different habitat types and uniformly distributed within the study area. 

The sampling effort, quantified by the number of investigated sample squares, increased from the first to the last session to follow the spread of the species throughout the area: from 2007 to 2012, we only investigated sample squares within the mountainous zone (*n* = 11); in 2015, we added the sample squares within the upper-hill zone (*n* = 9) and, in 2018, those within the lower-hill zone (in 2018: *n* = 9 + in 2020: *n* = 5; total: *n* = 14). In total, we had 34 sample squares (Table 1). The decision to add sample squares was prompted by the first evidence of occasional wolf presence in each respective zone.

During each survey, we walked the selected routes to record indirect signs of wolf presence, mainly corresponding to scats, footprints, ground scratchings, and feeding remains. We identified them by size, shape, and location [27]. Signs of wolf presence are almost always easily identifiable, especially in the absence of other wild large predators; nevertheless, scats and other signs of presence of uncertain origin were discarded. The adoption of the described method facilitated the detection of permanently occupied areas because most of the detected indirect signs of presence corresponded to scent markings (i.e., scats and ground scratchings) left by stable wolves within their territories [28,29]. Recorded data were georeferred with GPS (WGS 84 UTM zone 32N).

### 2.3. Occupancy Modelling

To model the range dynamics of the wolf, we used dynamic occupancy models [30], where *sites* corresponded to 34 sample squares of 25 km^2^. These models allowed us to study the species range dynamics by incorporating specific covariates to describe the parameters governing occupancy: the probability of occurrence during the 1st year of the survey (i.e., initial occupancy, ψ), the probability of colonization (γ), the probability of extinction (ε), and the probability of detection (p). We decided to include all sampling seasons in the analysis as (i) the ecological state of a site (whether a sample square is occupied or not) ideally remains unchanged through occasions (i.e., sampling seasons: winter, spring, summer, autumn) within the primary period (i.e., the year); (ii) we assumed that the majority of detected indirect signs of presence were most likely attributable to stable wolves rather than to dispersers [28,29]; (iii) dispersals have been documented during autumn, winter, and spring [31]; thus, excluding one season without having in-depth knowledge of the phenomenon in the study area would have been misleading. We excluded 2022 from the analysis, as we only had one sampling season.

We considered, as site-specific covariates, those variables that, according to the literature, are relevant to the species ecology and might influence its distribution. We used the land cover layers compiled by three Italian regions falling within the study area (Lombardy: https://www.geoportale.regione.lombardia.it; Piedmont: https://www.geoportale.piemonte.it; Emila-Romagna: https://geoportale.regione.emilia-romagna.it/; accessed on 15 October 2023) and summarized the reported detailed categories into broad classes mainly representing the key factors likely to influence wolf distribution: woodlands, open areas (e.g., grasslands, meadows, and pastures), uncultivated areas, arable lands, permanent crops, and urban areas. We also considered road networks made up of primary and secondary roads (https://www.openstreetmap.org; accessed on 15 October 2023). We considered, as yearly-site-level covariates, the estimated abundance of the two widespread wild ungulate species, roe deer and wild boar, in each monitoring session. Following the approach proposed by Sanz-Pérez et al. [32], we used wild ungulate occurrence data—which, in our case, were the indirect signs of presence identified in the field recorded during seasonal surveys—to delineate their utilization distributions (UDs) through kernel density estimation (KDE) [33]. Using a probabilistic approach, this method describes the intensity of the use of geographical space by animals, generating a surface that describes the probability of being at a particular location in any part of the study area. We used a fixed kernel estimator and applied the reference smoothing factor (h_ref_). Site-specific and yearly site-specific covariates were referred to each sample square composing the study area. Lastly, seasons and years were considered as observation-level covariates both as categorical variables to see if there was any trend in detection over time (Table 2). We checked potential multicollinearity among covariates using the variance inflation factor (VIF); we retained VIF = 3 as the threshold value [34,35]. We used the “usdm” package [36] in R [37].

We built models following a stepwise procedure: in the first model (the null model), we assumed that all four parameters (ψ, ɣ, ε, p) were constant across sites and surveys; no covariates were included in this model. Subsequently, we proceeded by fitting covariates for the detection parameter (p), and the best detection model structure was carried forward to estimate additional drivers of other parameters (ψ, ɣ, ε), which were modeled using the standardized (z-score) covariates. We followed an information–theoretic approach for comparing models containing different combinations of covariates for each of the four parameters (ψ, ɣ, ε, p). Model comparisons were made on the basis of AICc scores and Akaike weights (wi) [46]. Therefore, we used the best model(s) in the estimation of detection probability, initial occupancy, colonization, and extinction of wolf presence in relation to predictive covariates. We averaged coefficients for covariates in models with Δ AICc ≤ 2 [46]. We assessed the fit of our model using the MacKenzie–Bailey goodness-of-fit test based on bootstrapping (100 iterations). We performed dynamic occupancy models using the “unmarked” package [47] in R [37].

### 2.4. Diet

The wolf diet was studied from 2007 to 2022 through scat analysis. Scats were analyzed to identify the consumed items from undigested remains: hairs, bones, hoofs, and claws (medium- and large-sized mammals); hairs and mandibles (small mammals); and seeds and leaves (fruits and plants). Remains were identified by comparison to a private reference collection (hairs, mandibles, and seeds collected in the field from carcasses or plants) and an atlas [48,49,50]. We observed the hairs with an optical microscope (Leica DM750; Leica Microsystems, Wetzlar, Germany) to identify the consumed species from the characteristics of cortical scales, medulla, and roots. From undigested remains, the proportion of consumed items for each scat was calculated and then converted into a percent volumetric class, following the procedure used in the previous researches of the same project to allow the comparison of results [18,19,22,23]. We determined the diet composition for each monitoring session. To assess the adequacy of sample size, i.e., the number of analyzed scats, in describing the diet of wolves, we used the Brillouin index (Hb) [51]. Variations between monitoring sessions of the consumed categories were tested via nonparametric multivariate analysis of variance (NPMANOVA) using the post hoc Bonferroni correction of the p-values for pairwise comparisons [52]. Moreover, variations between monitoring sessions within categories were tested via nonparametric analysis of variance (Kruskal–Wallis test with Dunn test for pairwise comparisons). Lastly, we assessed wolf diet breadth in each monitoring session using a normalized Levins’ B index [53]. 

Diet analyses were performed by subdividing the samples based on the three elevational zones, i.e., mountainous, upper-hill, and lower-hill zones. 

## 3. Results

### 3.1. Range Dynamics

From 2007 to 2021, we confirmed the presence of wolves at 33 of 34 sites. The naive occupancies, that is, the proportion of sites where the wolf was detected at least once based on the conducted surveys, were relatively high in each monitoring session (mean: 0.70 ± SE 0.05; range: 0.40–0.91).

The VIF values of the considered covariates indicated that two (woodlands and road density) had serious multicollinearity; thus, we excluded them from the analyses. 

We obtained three models with a ΔAIC < 2; the obtained model set included redundant models; thus, we decided to consider the most parsimonious one [54]. This model included roe deer and wild boar abundances as covariates influencing colonization, urban areas cover as a covariate influencing extinction, and the year of monitoring as a covariate influencing detection (Table 3). The MacKenzie–Bailey goodness-of-fit test indicated that the model adequately fit the data (χ^2^ = 272.0, *p* = 0.08). 

The detection probability as a function of year had the strongest support. The effect of year on detection probability was negative from 2008 to 2011; overall, our model demonstrated high detectability (0.61 ± SE 0.08), although it was slightly lower from 2008 to 2011 (0.22–0.45) and increased thereafter (0.65–0.96; Table 3 and Figure 2).

The estimate of occupancy (0.74 ± SE 0.03) did not vary substantially from the naive occupancy because, as previously stated, the detection probability was relatively high; therefore, estimates are reasonably unbiased [30]. There was a major difference between the model-based estimated occupancy and the naive occupancy in 2009 (Table 3 and Figure 3).

The top model indicated that the range dynamics of the wolf were driven by prey abundance and human presence. The probability of colonization increased with roe deer abundance and decreased with wild boar abundance, whereas the probability of extinction increased with urban area cover (Table 3 and Figure 4).

The mountainous zone of the study area was the first to be occupied; in this zone, the predicted occupied sites increased from 2007 to 2011, when all sites had occupancy probabilities > 0.80. Starting from 2010, even sites within the upper-hill zone showed an increase in occupancy probabilities and became entirely occupied by 2018. The lower-hill zone was the last to be occupied, and a few sites remained with a low occupancy probability up to 2021. Overall, wolf occupancy in the Lombardy Apennines progressed from the mountainous to lower-hill zones gradually, with very limited extinction probability (Figure 5).

### 3.2. Diet

We analyzed 941 wolf scats (2007: 17; 2008: 10; 2009: 6; 2010: 51; 2011: 19; 2012: 51; 2015: 112; 2018: 94; 2019: 125; 2020: 203; 2021: 253); in some years, the sample size was not adequate to represent wolf diet as a consequence of the dissimilarities in sampling seasons (Table 1), so we considered, as subsamples, scats collected during the eight monitoring sessions (Table 1), putting together scats collected in the second and third monitoring sessions given the low number of scats collected from 2008 to 2009 (2007–2008: 27; 2009–2010: 57; 2011–2012: 70; 2015: 112; 2018–2019: 219; 2020: 203; 2021: 253).

Through scat analysis, we detected nine food categories consumed by the wolf: livestock, wild ungulates, small mammals, medium-sized mammals, birds, invertebrates, fruits, grasses, and garbage (Table 4).

The scats analyzed for each monitoring session were an adequate enough sample to represent the wolf diet in every considered zone, i.e., mountainous, upper-hill, and lower-hill zones (Appendix A).

In the mountainous zone, the wolf diet was significantly different between monitoring sessions (NPMANOVA: F = 17.10; *p* < 0.001); in particular, the diet of the first four monitoring sessions (2007–2008; 2009–2010; 2011–2012; 2015) differed from the diet of the last three monitoring sessions (2018–2019; 2020; 2021) (post hoc pair-wise comparisons with Bonferroni correction). The consumption of livestock increased from 2007–2008 to 2011–2012, then it rapidly decreased (Kruskal-Wallis test: H = 116.30; df = 6; *p* < 0.001); conversely, the consumption of wild ungulates, overall the most consumed category, increased from the first to last monitoring sessions (H = 69.01; df = 6; *p* < 0.001). The consumption of medium-sized mammals was particularly high in 2007–2008; then, it decreased (H = 29.86; df = 6; *p* < 0.001). Even the consumption of grasses differed between sessions, which was noticeable in 2007–2008 and 2018–2019 (H = 230.49; df = 6; *p* < 0.001) (Figure 6a; Appendix A). In the upper-hill and lower-hill zones, the wolf diet did not differ significantly among monitoring sessions; in both zones, the category of wild ungulates was the most consumed, while other categories were occasional (Figure 6b,c; Appendix A).

Among wild ungulates, the wild boar and the roe deer were the most consumed species. In the mountainous zone, wolf consumption of the different species of wild ungulates was significantly different between monitoring sessions (NPMANOVA: F = 26.44; *p* < 0.001); in particular, the diet of the first four monitoring sessions (2007–2008; 2009–2010; 2011–2012; 2015) differed from the diet of the last three monitoring sessions (2018–2019; 2020; 2021) (post hoc pair-wise comparisons with Bonferroni correction). The consumption of wild boar generally decreased from 2007–2008 to 2021 (H = 82.79; df = 6; *p* < 0.001); conversely, the consumption of roe deer generally increased from 2007–2008 to 2021 (H = 191.20; df = 6; *p* < 0.001). Even the consumption of red deer differed among sessions, being particularly high in 2011–2012 (H = 61.85; df = 6; *p* < 0.001) (Figure 7a; Appendix A). In the upper-hill and lower-hill zones, wolf consumption of the different species of wild ungulates did not differ significantly between monitoring sessions; in both zones, the most consumed species was the roe deer, followed by the wild boar (Figure 7b,c; Appendix A).

Wolf diet was narrow in every considered zone (0.22–0.12); in the mountainous zone it was slightly, but significantly, broader during the first three monitoring sessions and it decreased starting from 2015. No other significant difference emerged (Figure 8).

## 4. Discussion

This study describes the changes in wolf range and diet that occurred from 2007 to 2021 in an in-depth investigated study area localised in Italy along the Northern Apen-nines.

As predicted, our results showed a gradual increase in the wolf range during the study period from the mountain to the lower hills. A very similar pattern has been documented in another focal area for the study of the wolf in Italy, that is, Tuscany [55]. In the Lombard Apennines, species occupancy steadily increased from 2007 to 2012; starting in 2015, the rate of increase slowed, and occupancy estimates reached a plateau. This might be a consequence of the saturation of our study area; in other words, most of the territory became permanently occupied by resident wolves. As highlighted by Zanni et al. [56], range expansion suggested that wolves preferred mountains and, once this zone became mostly occupied, started to settle at lower elevations as well. Interestingly, occurrence data in the Po Plain started to be recorded in 2015, and became more and more frequent during subsequent years (Meriggi, unpublished data). The observed pattern, i.e., the plateau in occupancy estimates within the Lombard Apennines and the increasing wolf detections within the Po Plain, suggests that, around 2015, the Lombard Apennines started to act as a source area of dispersers through a density-dependent phenomenon, with individuals being forced to move to lower elevations and more anthropized areas to find available territories. Indeed, dispersers generally move through territories free from packs, where they can eventually settle [56]. Thus, for dispersers coming from the Lombard Apennines, the best chance to avoid encounters with resident wolves was represented by the few dispersal routes connecting the Northern Apennines to the Central Alps across the highly fragmented and urbanized Po Plain [8]. 

The expansion of the wolf population toward hills and plains has been documented during the same years (2009–2013) in the entire northern sector of the Apennines (Piedmont, Lombardy, Liguria, Emilia-Romagna, and Toscana regions), where the colonization of new territories in nearly urbanized areas started to be observed, while historical stable territories in remote mountainous areas did not show an increase in the number of individuals [12]. We observed a similar trend in our study area, with increasing colonization of new areas in nearly urbanized landscapes located in the lower-hill zone. This landscape is markedly different from the undisturbed mountainous zone, where wolves settled in the early stages of their recolonization process. In this zone, woodland cover, which provides optimal areas for dens and rendezvous sites [38,41,57,58], is dense and uniform, whereas in the lower hills, it is restricted to small patches along streams. Moreover, woodlands and other natural habitats (e.g., uncultivated lands and shrublands), which are predominant in the mountainous zone, represent suitable habitats for many wild prey. This zone of our study area also includes vast grasslands and pastures along the mountain ridges. Livestock, very abundant on mountainous pastures during the grazing season, represented an easily accessible form of prey for wolves during the first years of the study, as anti-predator measures for livestock protection were not yet implemented at the time [9,21].

The expansion of the wolf range from the mountains to the hills has most likely been supported by prey availability. Indeed, the most important observed effect of the considered covariates in our models pertains to the colonization parameter, which was significantly influenced by prey abundance. In particular, the probability of colonization was positively influenced by roe deer abundance and negatively influenced by wild boar abundance. We documented very similar trends in the relative abundance values of the two species in the three zones, with lower values during the first monitoring sessions and higher values during the last monitoring sessions, with a slight decrease in the most recent years. Despite this, our data suggested a general increase in both species across the study period (Appendix B). Both the roe deer and the wild boar are classically described as forest-dwelling species, as woodlands contain all the necessary resources for their survival and persistence, like access to cover and feeding sites (roe deer in [59] and references therein; wild boar [60,61]). However, the species are able to exhibit considerable plasticity in terms of habitat selection; over recent decades, they have colonized fragmented landscapes and agroecosystems all over Europe [62,63], where they have taken advantage of the presence of scattered natural (e.g., riparian vegetation, residual broad-leaved woodlands) and semi-natural (e.g., hedgerows and arboreal cultivations) vegetated patches (roe deer [64]). In our study area, we observed that they are less abundant in the mountains compared with the hills, where they successfully exploit a highly heterogenous landscape rich in ecotones with broad-leaved woodlands interposed with vineyards and annual crops; the ecotones are indeed important habitats for both species (roe deer [59]; wild boar [65]). Despite their similar trends and habitat plasticity, their estimated density distribution had opposite effects on wolf colonization probability. More specifically, the two ungulates showed differences in most frequented areas; in other words, the areas of high-density probabilities of roe deer and those of wild boar are different and not overlapping. Such differences are more marked in the hills of the Lombard Apennines because these zones are characterized by higher habitat heterogeneity if compared with the mountains. This result implies a different use of the space by the two ungulates and suggests specific feeding habits for wolves inhabiting the Lombard Apennines. Indeed, it has been documented that wolves show a complete local adaptation to the ecology of the most consumed ungulate species [66].

Regarding the extinction parameter, we observed the significant positive effect of urban areas. As reported, in the Lombard Apennines, built-up areas mainly correspond to small towns and villages and reach an important percentage of cover only in the most northern sample squares (Figure 1), where the probability of occupancy was low or irregular across years (Figure 6). Human presence has been considered for decades to be a factor that negatively influences wolf habitat suitability; recently, it has been argued that this factor has only delayed the wolf recolonization of highly human-dominated landscapes [55], suggesting that wolves are able to exploit even extremely modified landscapes thanks to their behavioral adaptability [67].

In future research, it will likely be necessary to incorporate factors related to climate change into colonization and extinction modeling. Climate change is expected to impact both natural habitats and human activities in agroecosystems, thereby influencing prey and, consequently, predator species.

The diet of wolves in the Lombard Apennines was characterized by the very high consumption of wild ungulates and the secondary consumption of livestock; the other food categories were a negligible fraction of the diet. This result is not surprising, as the study area hosts rich and abundant wild ungulate guilds [68,69,70]. Moreover, as previous studies of this same research project already showed [22,23,24], wild ungulate consumption increased during the study period, following the general trend in the wolf diet observed in Europe [9]. The significant decrease in livestock consumption could also be related to the enhancement of anti-predator measures for livestock protection and/or the general drop in extensive farming with free-grazing animals that has occurred in recent years.

The results regarding wolf consumption of different species of wild ungulates deserve more attention. Conversely, based on the general pattern observed in recent years in Italy, with the wild boar as the most consumed species [71], we found that the roe deer was the staple of the wolf diet in the Lombard Apennines. By focusing on results regarding the mountainous zone, which was monitored for the whole study period, we documented opposite consumption trends for these two widespread wild ungulate species. We observed a gradual shift from the prevalent consumption of wild boar (2007–2008 and 2011–2012) to the prevalent consumption of roe deer, which increased from 2015 onward. These results were unlikely to be related to differences in the abundance of the two species, as we recorded similar trends considering the relative abundance indices; rather, they might be related to a specific adaptation of the predator to the local ecology of the most consumed species [66]. The roe deer is generally considered difficult prey for wolves in wooded habitats, as a consequence of its behavior of living isolated or in small groups [72]. However, in the hills of the Lombard Apennines, this species has reached, in a few years, very high densities, increasing both the encounter and detection chances for wolves. Indeed, in these zones, where wolves have settled in most recent years, predator colonization has followed that of the most consumed species occupying areas with high-density probabilities. Interestingly, very recent studies carried out in Europe have recorded an opposite trend in wolf consumption of the same species, with an increasing predominance of the wild boar over the roe deer (e.g., Spain [73]; Poland [74]).

## 5. Conclusions

Based on our results, we can expect, in the near future, a remarkable expansion of wolves in the Po Plain, which currently represents the last free-from-packs area accessible for dispersers coming from the Apennines. This area hosts increasing roe deer sub-populations [75] despite the high human disturbance, corresponding to dense road networks, continuously built-up areas, and constant human presence. As many studies have pointed out, once prey species are available, wolves are able to settle and persist in any place if human persecution is low [43]. We expect, in particular, that wolf expansion will follow that of the roe deer across the Po Plain with the settlement of wolf packs in roe deer high-density areas. In this human-dominated landscape, new scenarios related to human–wolf conflicts are likely to become more pressing. Issues, such as the depredation of pets or the potential spread of zoonosis, may emerge, giving rise to new social conflicts.

## Figures and Tables

**Figure 1 animals-14-00735-f001:**
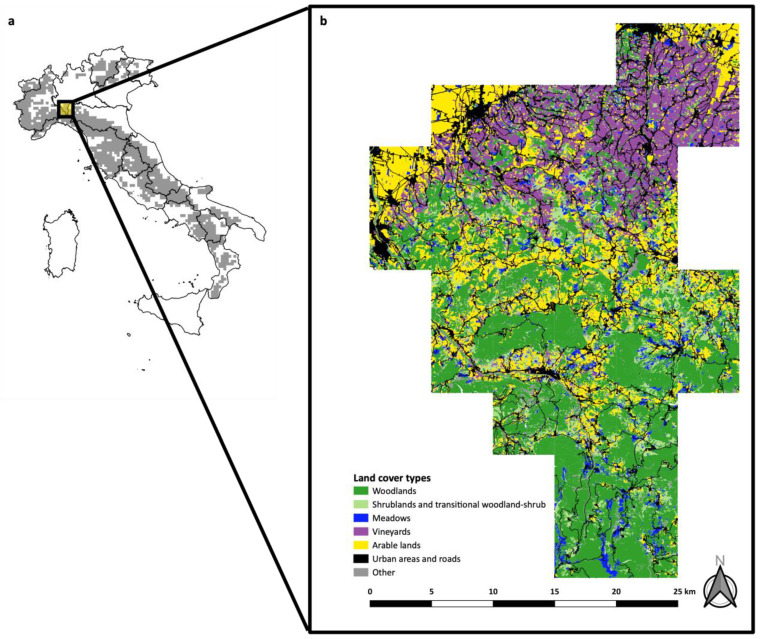
The study area: (**a**) location of the Lombard Apennines in Italy and along the wolf range (in gray) obtained from the report on the species of community interest (2013–2018) and updated with more recent data (2022); (**b**) land cover classes mainly characterizing the landscape of the study area.

**Figure 2 animals-14-00735-f002:**
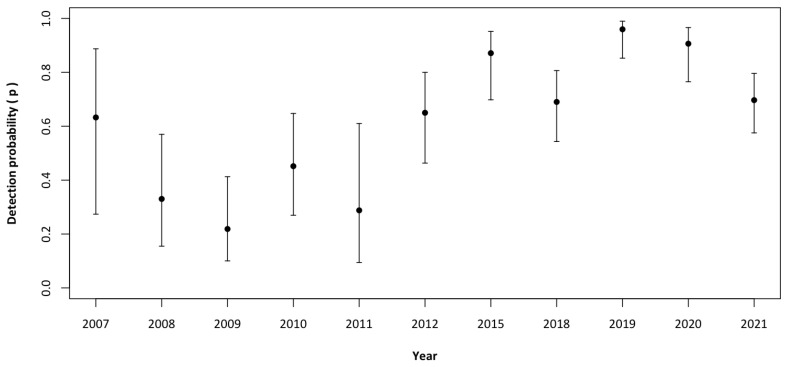
Estimated mean probability of detection (±95% CI) of the wolf from 2007 to 2021 in the Lombard Apennines (Northern Apennines, Italy).

**Figure 3 animals-14-00735-f003:**
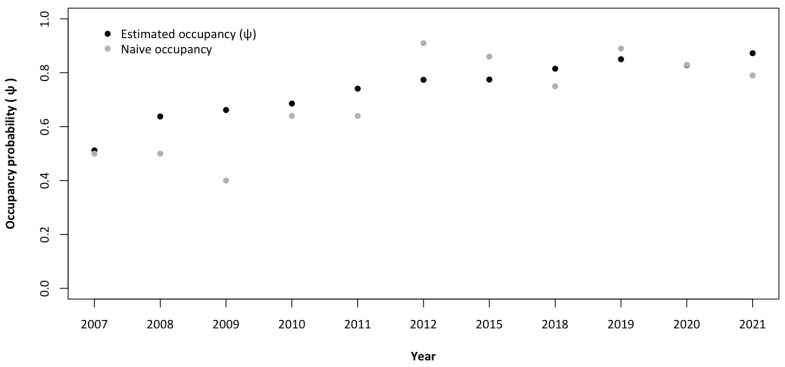
Estimated mean probability of occupancy (±95% CI) and naive occupancy of the wolf from 2007 to 2021 in the Lombard Apennines (Northern Apennines, Italy).

**Figure 4 animals-14-00735-f004:**
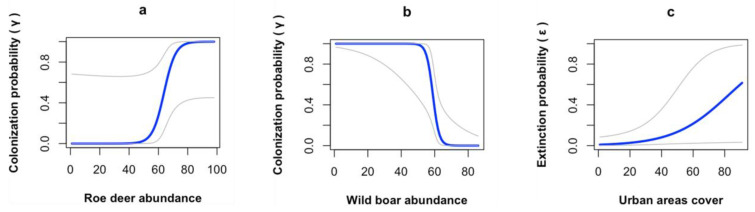
Response curves of colonization probability to (**a**) roe deer abundance and (**b**) wild boar abundance and response curve of extinction probability to (**c**) urban area cover estimated based on the top model obtained for the wolf in the Lombard Apennines (Northern Apennines, Italy). Gray lines depict lower and upper 95% CIs.

**Figure 5 animals-14-00735-f005:**
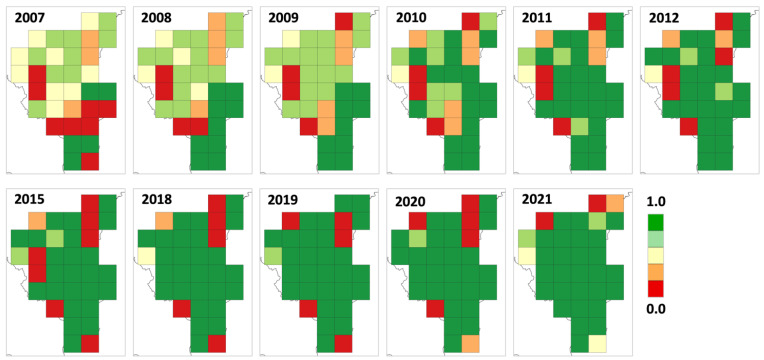
Maps of the estimated occupancy probabilities for the wolf from 2007 to 2021 in the Lombard Apennines (Northern Apennines, Italy).

**Figure 6 animals-14-00735-f006:**
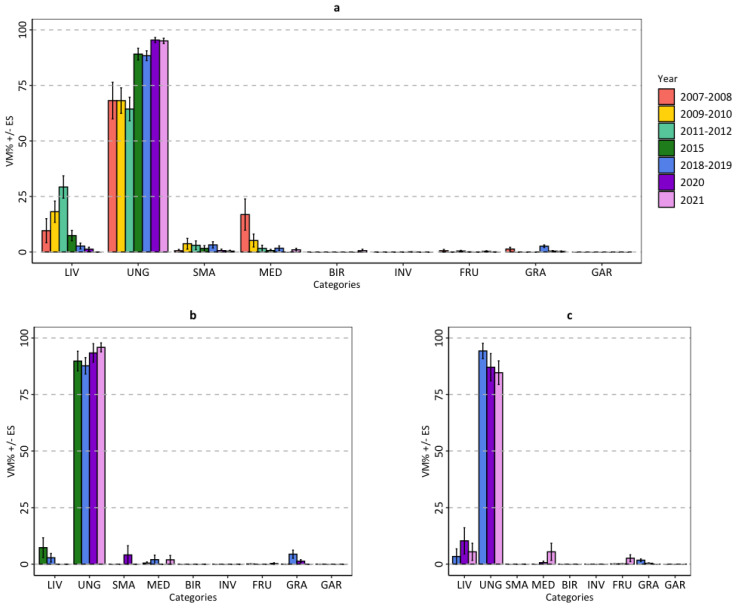
Food habits of the wolf in the (**a**) mountainous, (**b**) upper-hill, and (**c**) lower-hill zones of the Lombard Apennines (Northern Apennines, Italy) expressed as mean percent volume (VM% ± SE) of consumed categories: liv = livestock; ung = wild ungulates; med = medium-sized mammals; sma = small mammals; bir = birds; inv = invertebrates; fru = fruits; gra = grasses; gar = garbage.

**Figure 7 animals-14-00735-f007:**
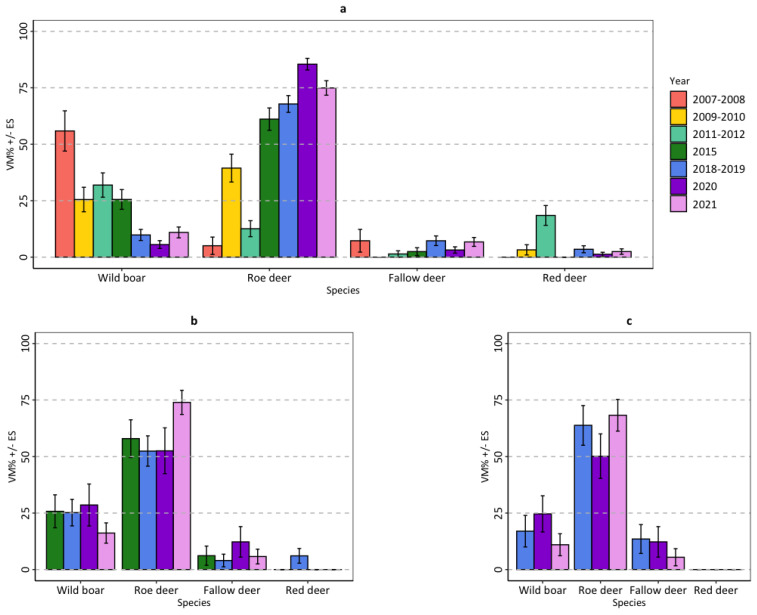
Food habits of the wolf in the (**a**) mountainous, (**b**) upper-hill, and (**c**) lower-hill zones of the Lombard Apennines (Northern Apennines, Italy) expressed as mean percent volume (VM% ± SE) of consumed wild ungulate species.

**Figure 8 animals-14-00735-f008:**
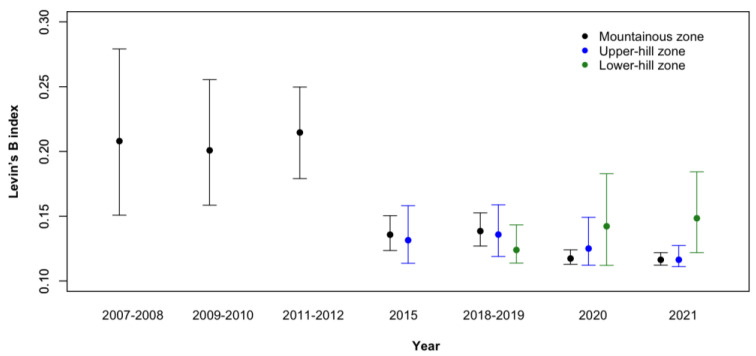
Breadth of the wolf diet in the three zones of the Lombard Apennines (Northern Apennines, Italy) measured by Levins’ B index. Error bars depict lower and upper 95% CIs.

**Table 1 animals-14-00735-t001:** The sampling effort for wolf monitoring in the Lombard Apennines (Northern Apennines, Italy) from 2007 to 2022.

Monitoring Session	Year	Seasonal Surveys ^1^				N. of Surveyed Sample Squares	N. of Sampling Routes	Length of Routes (Mean ± SD)	Total Length of Routes
1	2007	w	sp	**su**	**a**	11	12	6.8 ± 2.5 km	81.9 km
	2008	**w**	**sp**	su	a				
2	2008	w	sp	su	**a**	11	12	6.8 ± 2.5 km	81.9 km
	2009	**w**	**sp**	**su**	a				
3	2010	w	**sp**	**su**	**a**	11	13	6.8 ± 2.3 km	87.9 km
	2011	**w**	sp	su	a				
4	2011	w	sp	su	**a**	11	13	6.8 ± 2.3 km	87.9 km
	2012	**w**	**sp**	**su**	a				
5	2015	w	**sp**	**su**	a	20	26	5.4 ± 2.3 km	139.2 km
6	2018	w	sp	**su**	**a**	29	37	5.4 ± 2.6 km	199.1 km
	2019	**w**	**sp**	su	a				
7	2020	**w**	sp	**su**	a	34	39	5.3 ± 2.6 km	205.8 km
8	2021	w	**sp**	**su**	**a**	34	42	5.1 ± 2.6 km	215.0 km
	2022	**w**	sp	su	a				

^1^ In bold, the sampling seasons within each year (w = winter, from December to February; sp = spring, from March to May; su = summer, from June to August; a = autumn, from September to November).

**Table 2 animals-14-00735-t002:** Description and expected effects of covariates used to describe the range dynamics of the wolf in the Lombard Apennines (Northern Apennines, Italy).

Covariate	Description	Parameter	Expected Effect	References
Site level				
Woodlands	Percentage cover of mixed, coniferous, and broad-leaved woodlands	Initial occupancy (Ψ)	+	[38,39,40,41]
Open areas	Percentage cover of open areas (grasslands, meadows, and pastures)	Initial occupancy (Ψ)	+	[18,42]
Uncultivated areas	Percentage cover of unutilized areas	Initial occupancy (Ψ)	+	[18]
Arable lands	Percentage cover of cultivated fields	Initial occupancy (Ψ)	−	[8]
Permanent crops	Percentage cover of vineyards and fruit trees	Initial occupancy (Ψ)	−	[8]
Urban areas	Percentage cover of built-up areas	Extinction (ε)	+	[43,44,45]
Road density	Length of primary and secondary roads (km/km^2^)	Extinction (ε)	+	[43,44,45]
Yearly site-level				
Roe deer abundance	KDE based on occurrence data recorded during seasonal surveys	Colonization (γ)	+	[40]
Wild boar abundance	KDE based on occurrence data recorded during seasonal surveys	Colonization (γ)	+	[40]
Observation level				
Year	Year of monitoring	Detection (p)	+	
Season	Season of survey	Detection (p)	−/+	

**Table 3 animals-14-00735-t003:** Estimated regression coefficient (β) and associated standard error (SE) values for occupancy (ψ), detection (p), colonization (γ), and extinction (ε) from the top model.

Parameter	Term	β	SE
Occupancy	Intercept	0.04	0.68
Colonization	Intercept	1.63	1.21
	Roe deer abundance	5.09	2.66
	Wild boar abundance	−10.67	4.76
Extinction	Intercept	−3.64	0.91
	Urban areas	1.14	0.56
Detection	Intercept	0.55	0.77
	Y 2008	−1.25	0.92
	Y 2009	−1.82	0.90
	Y 2010	−0.74	0.87
	Y 2011	−1.45	1.04
	Y 2012	0.07	0.87
	Y 2015	1.36	0.94
	Y 2018	0.25	0.84
	Y 2019	2.63	1.06
	Y 2020	1.72	0.95
	Y 2021	0.29	0.82

**Table 4 animals-14-00735-t004:** Food habits of the wolf in the (a) mountainous (b) upper-hill, and (c) lower-hill zones of the Lombard Apennines (Northern Apennines, Italy), expressed as mean percent volume (VM% ± SE).

Categories and Species	Mountain (*n* = 696)		Upper Hills (*n* = 156)		Lower Hills (*n* = 89)	
	VM%	SE	VM%	SE	VM%	SE
**Livestock**	**6.55**	**0.88**	**2.39**	**1.09**	**6.08**	**2.44**
*Ovis aries*	1.04	0.32	0.87	0.67	2.07	1.46
*Capra hircus*	2.22	0.52	0.73	0.64	1.81	1.30
*Bos taurus*	3.01	0.61	0.79	0.60	2.20	1.55
*Equus caballus*	0.29	0.20	0.00	0.00	0.00	0.00
**Wild ungulates**	**86.70**	**1.11**	**91.68**	**1.72**	**88.44**	**2.90**
*Sus scrofa*	16.33	1.34	22.88	3.13	16.62	3.71
*Capreolus capreolus*	62.22	1.74	60.61	3.62	61.91	4.83
*Dama dama*	4.45	0.77	6.28	1.93	9.91	3.15
*Cervus elaphus*	3.71	0.70	1.90	1.04	0.00	0.00
**Small mammals**	**1.70**	**0.46**	**0.63**	**0.63**	**0.00**	**0.00**
**Medium-sized mammals**	**1.90**	**0.49**	**1.36**	**0.89**	**2.38**	**1.56**
**Birds**	**0.14**	**0.14**	**0.00**	**0.00**	**0.00**	**0.00**
**Invertebrates**	**0.01**	**0.01**	**0.00**	**0.00**	**0.00**	**0.00**
**Fruits**	**0.15**	**0.07**	**0.12**	**0.10**	**1.12**	**0.62**
**Grasses**	**0.72**	**0.15**	**1.60**	**0.59**	**0.67**	**0.20**
**Garbage**	**0.004**	**0.004**	**0.003**	**0.003**	**0.00**	**0.00**

## Data Availability

The data are available upon reasonable request from the corresponding author.

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
