# Peer review of "Changes in Wolf Occupancy and Feeding Habits in the Northern Apennines: Results of Long-Term Predator–Prey Monitoring"

_animals, 2024, doi:10.3390/ani14050735_

Round 1
Reviewer 1 Report
Comments and Suggestions for Authors
The authors present valuable work on changes in wolf occupancy and feeding habits during a long-term monitoring period. I consider they must provide additional information on the methodology used and clarify some doubts regarding the statistical analysis performed. My main concerns can be easily addressed; therefore, I truly believe that the authors can do a proper review of their work to improve it and make this manuscript suitable for publication in Animals. Please see my specific comments below.
Abstract
Page 1. L 32-34. Please clarify this statement as if the prevalent prey was the wild boar in 2007-2008, it is weird that the roe deer was from 2007 and onwards, it should be at least from 2008 or maybe 2009?
Consider including here the scientific names for both the wild boar and roe deer.
Introduction
Page 2. L81-88. The first statement looks like a result just before stating the aims in the next paragraph. Consider rewriting or moving this paragraph to the material and method section. The last sentence (L88-89) can be kept in the introduction.
Page 2. L96-97. A lower or a higher magnitude? Please specify.
Page 3. L100-101. As this is a prediction, the concrete ungulate species that is predicted to be the most consumed should be mentioned, and also add whether this prediction is supported by a higher abundance/density of this species in the study areas.
Material & Methods
Page 3. L134-135. Why there is no data for other years during the study period? It would be interesting to know the trends in livestock numbers as this can be an additional modulating factor. See a comment in the Discussion section.
Page 3. L137. If related, consider stating the same temporal range as on Page 1, L75.
Page 4. L154. Consider including here the total number of sample squares.
Page 4. L158-159. Table 1. There is no data for 2013, 2014, 2016 and 2017. Please include it or explain somewhere in the text the lack of data for this period and the main reason. In addition, the authors present data for winter 2022. Was the data from this year also used? Also in Table 1, consider presenting data for sample squares and routes in a different way (e.g., by splitting the table into rows) to clearly state the number for each year or period.
Page 5. L178-183. If dispersals have been documented almost throughout all seasons, why are the indirect signs found attributable to stable wolves?
Page 5. L193-195. Mean values of the abundance for each species could be included here or somewhere in the text.
Page 6. L205. Table 2. Is “initial occupancy” the same as “probability of occurrence”? stated on Page 5. L174? In that case, I would suggest using the same terminology throughout the manuscript to facilitate the reading.
Page 7. L226-227. Please state whether the collection is private or in a public repository/museum. Please also reference the mentioned atlas.
Page 7. L234. Include a reference for the Hb index.
Page 7. L241-246. Could this design, i.e., including additional zones throughout the years, have influenced the results?
Page 12. L333-346. Did the authors find any change in the most consumed prey species among seasons?
Results
Page 7. L253-254. Were both covariates excluded? If correlated variables are removed this can cause another problem as if these variables eliminated from the model are significant, we will be omitting a relevant variable, which may cause the estimators of the coefficients of the model and its variance to be biased to other components. Considering the ecological importance of both woodlands and road density for the species (Page 5. L184-193, but also along the discussion section woodlands relevance is highlighted), have the authors tried any other approach to address this collinearity issues instead of excluding both variables from the analyses?
Page 8. Table 3. Concerning my comment above on the sampling design, coincidentally, detection values seem to increase in and/or after those years when additional zones were added. This should be controlled somehow or at least discussed.
Page 8. Figure 2. Considering the lack of data for some years, I would suggest removing 14 years from the figure legend. The same for Figure 3.
Page 9. L290-293. Was there any available data on wolf occupancy of these lower-hill zones before being surveyed in 2018? This result might be a consequence of adding lower-hill zones from 2018 onwards, i.e., being the results of extending the study area (and lack of previous data for that zone) instead of wolves increasing occupancy areas. Have the authors controlled the effect of the sampling design?
Page 9. L293. Please check whether Figure 6 is correctly referenced at this point.
Discussion
Overall, the results are well-discussed according to the available literature. I would suggest the author consider my comments above on different aspects (especially on the occupancy results) and include here some brief explanations/justifications accordingly.
Page 14. L433. I would state here something about the shift in livestock consumption found from the first to the last surveyed years, adding the potential explanation stated in lines 395-397 or, alternatively, whether livestock practices have been minimized in the study area throughout the years, and this could have partially conditioned this diet trend.
Page 15. L448-457. Returning to my comment above in the Results section asking about the existence of differences in the main prey consumed (i.e., roe deer or wild boar) between seasons, if this occurred, it should be stated and discussed accordingly. As the authors argue that this was not a result of differences in roe deer and wild boar densities in the study area, an alternative explanation could be that wolf prey preference may be linked to prey vulnerability. See for example:
Guimarães, N.F.; Álvares, F.; Ďurová, J.; Urban, P.; Bučko, J.; Iľko, T.; Brndiar, J.; Štofik, J.; Pataky, T.; Barančeková, M.; et al. What Drives Wolf Preference towards Wild Ungulates? Insights from a Multi-Prey System in the Slovak Carpathians. PLoS ONE 2022, 17, e0265386. https://doi.org/10.1371/journal.pone.0265386
Author Response
The authors present valuable work on changes in wolf occupancy and feeding habits during a long-term monitoring period. I consider they must provide additional information on the methodology used and clarify some doubts regarding the statistical analysis performed. My main concerns can be easily addressed; therefore, I truly believe that the authors can do a proper review of their work to improve it and make this manuscript suitable for publication in Animals. Please see my specific comments below.
Abstract
Page 1. L 32-34. Please clarify this statement as if the prevalent prey was the wild boar in 2007-2008, it is weird that the roe deer was from 2007 and onwards, it should be at least from 2008 or maybe 2009?
Consider including here the scientific names for both the wild boar and roe deer.
We better specified the study period during which roe deer consumption definitively increased. Moreover, we added the scientific names for both the wild boar and the roe deer.
Introduction
Page 2. L81-88. The first statement looks like a result just before stating the aims in the next paragraph. Consider rewriting or moving this paragraph to the material and method section. The last sentence (L88-89) can be kept in the introduction.
We omitted the initial portion of the sentence, as it might have sounded like a preview of the results.
Page 2. L96-97. A lower or a higher magnitude? Please specify.
We opted to retain the original sentence, which implies a general consideration of the level of human pressure without specifying its magnitude. This choice maintains a neutral description.
Page 3. L100-101. As this is a prediction, the concrete ungulate species that is predicted to be the most consumed should be mentioned, and also add whether this prediction is supported by a higher abundance/density of this species in the study areas.
We rephrased iv prediction accordingly (L 105-108).
Material & Methods
Page 3. L134-135. Why there is no data for other years during the study period? It would be interesting to know the trends in livestock numbers as this can be an additional modulating factor. See a comment in the Discussion section.
We incorporated the suggested (available) data on livestock numbers and trends (L 142-144).
Page 3. L137. If related, consider stating the same temporal range as on Page 1, L75.
We changed accordingly.
Page 4. L154. Consider including here the total number of sample squares.
We shifted the sentence here as suggested (L 169-173).
Page 4. L158-159. Table 1. There is no data for 2013, 2014, 2016 and 2017. Please include it or explain somewhere in the text the lack of data for this period and the main reason. In addition, the authors present data for winter 2022. Was the data from this year also used? Also in Table 1, consider presenting data for sample squares and routes in a different way (e.g., by splitting the table into rows) to clearly state the number for each year or period.
We added the lacking years (2013, 2014, 2016, 2017) in the text (L 161-162). We changed (here and throughout the MS, when pertinent) the final year of the study because, as highlighted, we considered the data collected up to February 2022.
We tried to improve the readability of Table 1 splitting, as suggested, the table into rows, which corresponded to our monitoring sessions.
Page 5. L178-183. If dispersals have been documented almost throughout all seasons, why are the indirect signs found attributable to stable wolves?
We are confident that the majority of the indirect signs of presence we detected in our study area belonged to stable wolves because they were primarily scats, frequently found in predictable locations with multiple samples. These scats likely represented marking signs. Wolf marking behaviour is a crucial aspect of species territory establishment; in other words, scent marking, through scats deposition, is used for establish and maintain territories. Some authors have noted that the sampling method based on the detection of indirect signs of presence is particularly effective for studying stable wolves (packs), which engage in frequent and systematic territorial marking (Llaneza et al. 2014; Zub et al. 2003).
Furthermore, Mech and Boitani (2003) reported that dispersers tend to limit scent-marking in areas already occupied by packs. This suggests that the prevalence of scats, a common indirect sign of presence in our study, aligns with the marking behaviour of stable wolves rather than dispersers.
Page 5. L193-195. Mean values of the abundance for each species could be included here or somewhere in the text.
We reported mean annual values of the relative abundance index (IKA) for both species in Appendix A.
Page 6. L205. Table 2. Is “initial occupancy” the same as “probability of occurrence”? stated on Page 5. L174? In that case, I would suggest using the same terminology throughout the manuscript to facilitate the reading.
Yes, we specified it in the text to avoid confusion.
Page 7. L226-227. Please state whether the collection is private or in a public repository/museum. Please also reference the mentioned atlas.
Ok, we specified that the reference collection was private and that the samples were collected from carcasses or known plants (L 246-248).
Page 7. L234. Include a reference for the Hb index.
Ok.
Page 7. L241-246. Could this design, i.e., including additional zones throughout the years, have influenced the results?
No; we carried out the initial analyses considering the whole sample (i.e. without elevational zones) and we obtained the same general trend (increasing consumption of wild ungulates vs. decreasing consumption of livestock and increasing consumption of roe deer vs. decreasing consumption of wild boar). Thus, we decided to split the sample according to elevational zones, which reflected the sequential phases of monitoring. We decided to keep the subsamples as obtained results were more informative compared to the initial ones.
Page 12. L333-346. Did the authors find any change in the most consumed prey species among seasons?
Overall, we noted minimal seasonal variations in the consumption of the two primary prey species, roe deer and wild boar. These differences appear more stochastic than causative, considering the absence of detected seasonal trends.
Results
Page 7. L253-254. Were both covariates excluded? If correlated variables are removed this can cause another problem as if these variables eliminated from the model are significant, we will be omitting a relevant variable, which may cause the estimators of the coefficients of the model and its variance to be biased to other components. Considering the ecological importance of both woodlands and road density for the species (Page 5. L184-193, but also along the discussion section woodlands relevance is highlighted), have the authors tried any other approach to address this collinearity issues instead of excluding both variables from the analyses?
We assessed the significance of the two covariates running alternative models. Woodlands did not emerge as a crucial covariate in any model, while road density exhibited a very similar effect to urban areas. Due to the absence of collinearity issues, we chose to include urban areas as a covariate in our final analysis.
Page 8. Table 3. Concerning my comment above on the sampling design, coincidentally, detection values seem to increase in and/or after those years when additional zones were added. This should be controlled somehow or at least discussed.
Thanks for this useful consideration. Detection probability in dynamic occupancy models can be affected by various factors such as survey effort, sampling methodology, environmental conditions, and the behavior of the species. The increase of the survey effort, in particular, leading to a more thorough exploration of the study area, potentially results in higher detection probability.
Thus, we included this expected effect (i.e., an increase in detection probability throughout the study period) in Table 1.
Page 8. Figure 2. Considering the lack of data for some years, I would suggest removing 14 years from the figure legend. The same for Figure 3.
Ok.
Page 9. L290-293. Was there any available data on wolf occupancy of these lower-hill zones before being surveyed in 2018? This result might be a consequence of adding lower-hill zones from 2018 onwards, i.e., being the results of extending the study area (and lack of previous data for that zone) instead of wolves increasing occupancy areas. Have the authors controlled the effect of the sampling design?
This is a key point. In this paper we only reported data collected through our standardized monitoring protocol (i.e. the only suitable for the data analysis we carried out), but we also took note of occasional observations collected in our study area. This independent dataset was very informative and we used it to plan the steps of our research. In details, it mainly consisted of occasional observations (e.g. camera-traps or direct videos recorded by volunteers) which were scattered in space and time. When we started to collect observations in the lower-hills (2017-2018), we planned to extend the monitoring in this zone. These first observations concerned single wolves (not packs), which probably were dispersers. Indeed, dispersers are usually the first to appear in highly fragmented landscapes, such as the agro-ecosystem characterizing the lower-hills of the Lombard Apennines. Thus, as our standardized monitoring protocol facilitated the detection of permanently occupied areas, because most of the detected indirect signs of presence corresponded to scent markings (i.e. scats and ground scratchings) left by stable wolves within their territories, our results regarding wolf colonization of the area should be reasonable.
We added a short explanation in the text (L 174-175).
Page 9. L293. Please check whether Figure 6 is correctly referenced at this point.
Thanks, Figure 5 was the right one.
Discussion
Overall, the results are well-discussed according to the available literature. I would suggest the author consider my comments above on different aspects (especially on the occupancy results) and include here some brief explanations/justifications accordingly.
Page 14. L433. I would state here something about the shift in livestock consumption found from the first to the last surveyed years, adding the potential explanation stated in lines 395-397 or, alternatively, whether livestock practices have been minimized in the study area throughout the years, and this could have partially conditioned this diet trend.
Thanks for this suggestion; we added it in our Discussion (L 469-471).
Page 15. L448-457. Returning to my comment above in the Results section asking about the existence of differences in the main prey consumed (i.e., roe deer or wild boar) between seasons, if this occurred, it should be stated and discussed accordingly. As the authors argue that this was not a result of differences in roe deer and wild boar densities in the study area, an alternative explanation could be that wolf prey preference may be linked to prey vulnerability. See for example:
Guimarães, N.F.; Álvares, F.; Ďurová, J.; Urban, P.; Bučko, J.; Iľko, T.; Brndiar, J.; Štofik, J.; Pataky, T.; Barančeková, M.; et al. What Drives Wolf Preference towards Wild Ungulates? Insights from a Multi-Prey System in the Slovak Carpathians. PLoS ONE 2022, 17, e0265386. https://doi.org/10.1371/journal.pone.0265386
Thanks for the suggested paper; as stated, we did not detect important seasonal trends in the consumption of wild ungulates. Roe deer and wild boar are abundant and widespread in our study area; we detected similar annual trends in their abundance (please see the IKA plots in Appendix A) and there are no external factors potentially influencing their vulnerability to predation (e.g. the snow cover is restricted to few weeks or days during the winter and it is limited to the mountain ridges).
Reviewer 2 Report
Comments and Suggestions for Authors
Dear Authors,
The manuscript is describing an interesting and important topic. It is a well-planned study reported in a good structure. The approach is adequate. However, I feel that this version still requires improvements in the formulation of predictions and in the description of the methods.
Please, check my comments in the uploaded pdf file.

Author Response
Dear Authors,
The manuscript is describing an interesting and important topic. It is a well-planned study reported in a good structure. The approach is adequate. However, I feel that this version still requires improvements in the formulation of predictions and in the description of the methods.
Please, check my comments in the uploaded pdf file.
L 11-13: It is not clear, how long period was covered by this monitoring during which those changes has occurred. Moreover, I would add some words about the methods, since it is not clear what kind of monitoring was this.
We added the suggested details (i.e. study period and used sampling method).
L 22-23: Latin name? Or at least to write grey wolf would be good to write.
We added the scientific name.
L 24-25: This sentence was missing for me above.
We modified the sentence in the Simple Summary accordingly, but we kept this sentence here as it gives more details on the used sampling protocol.
L 31: That is somehow strange this finding that wolf presence decreased by wild boar abundance, mainly without any explanation about the casual relationship. Based on the next sentence it is not due to wild boar but it is the consequence of preference for roe deer.
We changed the Abstract removing the part related to wild boar, because, as highlight by the reviewer, without any explanation, it could have been misleading.
Following reviewer’s next suggestions, we tried to improve the explanation of our results related to wild boar effect on wolf colonization in the Discussion.
L 58: Point is lacking from the end of the sentence.
We added the point.
L 98-99: I do not like those iii and iv predictions in this form, since now it is not well connected to the wolf area expansion and the food supply as the potential cause of it. Based on the abstract I expected predictions related to roe deer vs. wild boar prey choice not to livestock vs. wild ungulate food selection. Moreover, until now nothing was about diet breadth, so it is not introduced, it is arriving from nowhere.
Otherwise, I do not understand why did not you expect an increasing amount of livestock (and not wild ungulate) in the diet if wolf has been spreading towards human dominated landscape. You should state that there are less livestock there than in the mountainous areas, if it is the case.
We rephrased iii and iv predictions based on these valuable suggestions (L 103-108).
L 115: Put comma between surface and respectively.
We added the comma.
L 127: Human population.
We changed accordingly.
L 149: It is not clear what is the meaning of one session, why it is 8? You should define it, i.e. all seasons included once, I guess according to the Table.
We explained what we meant for monitoring session (i.e. a survey period made up by seasonal repetitions of the routes) (L 158-160); moreover, we explained that the monitoring sessions are 8 because in some years we were not able to carry out the data collection (L 161-162).
And how it is related to the years in the data analysis?
Regarding the occupancy modelling, we considered the seasons as secondary periods and the years as primary periods. We better specified it (L 197-198 and 202-203). As for the diet analysis, we considered the 8 monitoring sessions.
Table 1: It is a bit hard to understand in all cases that the values in the last 4 columns are related to which years and seasons. They should be always positioned in the middle or the beginning of the related sessions.
We tried to improve the readability of Table 1 splitting the table into rows, which corresponded to our monitoring sessions.
Table 1: In the abstract and later in the text you always mention 2007-2021, but it is 2022.
We changed (here and throughout the MS) the final year of the study because, as highlighted, we considered the data collected up to February 2022.
L 163-164: I guess DNA identification could only be a certain technique for recognition and to distinguish for example from scats of feral or stray dogs.
There aren’t feral or stray dogs in our study area.
L 188: Did you consider the proportion of those habitat types within the sample squares? Probably, but not stated.
Yes, we specified it in the text.
L 191: What is the meaning of considered road network? The total length per square was used?
We specified it in Table 2: length of primary and secondary roads (km) / area of sample square (km2).
L 196: How their abundance was given? I recommend to provide the description of the determination of ungulate species density, not just to give a citation about the whole related methodology, since it is a crucial point and the basis of one of the main results. So what kind of data and how they were collected in the field?
Ok, we specified the data we used (L 216-217). We considered as occurrence data the indirect signs of presence attributed to wild boar or roe deer recorded along the sampling routes during the seasonal surveys. We also added the details regarding the KDE (L 221-223).
L 246: It is not clear. Was not there scat collection in each year of the monitoring only 2007, 2008, 2009, 2010, 2011, 2012, 2015, 2018 and 2020? No collection in 2019, 2021 and 2022? You should state about it in the methods.
We collected wolf scats during each monitoring session and, within sessions, during each season. No missing data.
Also it is not clear how in what proportion the different zones were included in the collected sample in the different years, how the different zones were represented in the sample? Does it mean 10-10-10 cells of the 3 zones or something like that?
We have 11 squares in the mountainous zone, 9 squares in the upper-hill zone, and 14 squares in the lower-hill zone. In total, we had 34 sample squares (L 169-174).
Figure 2: Meaning of session is not defined.
We added the suggested definition in 2.2 Data collection (L 158-160).
L 279: Difference between estimated occupancy and naive occupancy has not been well defined yet, neither in the Methods. It can be confusing.
A simple definition of naïve occupancy is reported in 3.1 Range dynamics (L 269-271).
L 419: I do not agree with this statement that wild boar density decrease has a positive effect on wolf colonization. How this kind of effect could work in reality? Wolf avoids wild boar inhabited areas? It is not a direct effect but it can be a correlation.
We agree with the reviewer: we did not suggest that wolves avoided wild boar areas, but we only observed that wolf colonization of previously unoccupied areas was driven by roe deer abundance.
Roe deer and wild boar are widespread and abundant in our study area, but they showed differences in most frequented areas (i.e. the areas with high probabilities of having higher density of roe deer and those of wild boar are different and not overlapped). Such differences are probably related to habitat and more marked in the hills of the Lombard Apennines, because these zones are characterized by higher habitat heterogeneity if compared to the mountains, where woodlands are predominant and uniformly distributed. In the hills, where wolves settled in most recent years, predator colonization followed that of its most consumed species, which was the roe deer, occupying the areas with high probabilities of having higher density of roe deer.
We tried to improve the explanation of our results in the Discussion (L 441-446 and L 486-489).
Reviewer 3 Report
Comments and Suggestions for Authors
In the present work " Changes in wolf occupancy and feeding habits in the Northern Apennines: results of a long-term predator-prey monitoring", Authors model wolf range dynamics, using dynamic occupancy models, considering land cover types and wild ungulate abundances as covariates.
The manuscript is well written and structured. Anyway, some little changes have to be performed in order to go ongoing with the publication. If Authors will follow the suggestion given, I will certainly recommand this paper for the publication.
Introduction. Line 88. You speak about the possible factors which have influ-enced the recent changes in wolf range and diet and the fact that the present work may help us to undestand this aspect. Anyway, Authors should give some suggestion referring to the situation, also regarding the literature already published. This past may be introducted here and explained more in the discussion, but should be cited here first.
Figure 4. I suggest you to use two different colors to represent the curve, because black and grey may create confusion, expecially in the overlap area.
Discussion. Line 395. Authors speack about the abudant presence of livestock in mountain area and the role of prey where anti-predator measures for livestock protection were not yet implemented at the time. Did you think that is just a consequence of mismanagement? Or there is also a lack of knowlegde and poor communication from the competent authorities? I think that you must underline this aspect.
Line 409-412. All these information about the environmental conditions now, with the development of new technologies such as GIS and Remote Sensing, represent an incredible source of data. Not only for research interests but also for the availability of health information (for example about zoonoses in wildlife) and the important impact to Public Health (https://doi.org/10.3390/rs12213542).
Line 414-423. In this part, I advice you also to due a brief consideration to climate change that represent an important factor impactful on the habitat conditions that reflecting on prey presence and, consequently of wolf presence (https://doi.org/10.1007/s00442-017-4017-y).
Conclusion. What will be the future challenges in the scenario you have proposed? The research will be able to answer to new human-livestock-wildlife interface questions?
Author Response
In the present work "Changes in wolf occupancy and feeding habits in the Northern Apennines: results of a long-term predator-prey monitoring", Authors model wolf range dynamics, using dynamic occupancy models, considering land cover types and wild ungulate abundances as covariates.
The manuscript is well written and structured. Anyway, some little changes have to be performed in order to go ongoing with the publication. If Authors will follow the suggestion given, I will certainly recommend this paper for the publication.
Introduction. Line 88. You speak about the possible factors which have influenced the recent changes in wolf range and diet and the fact that the present work may help us to understand this aspect. Anyway, Authors should give some suggestion referring to the situation, also regarding the literature already published. This past may be introduced here and explained more in the discussion, but should be cited here first.
Ok, we added some suggestions regarding the mentioned factors potentially influencing wolf expansion (L 90-93).
Figure 4. I suggest you to use two different colors to represent the curve, because black and grey may create confusion, especially in the overlap area.
Ok, we changed the colors of the plotted curves.
Discussion. Line 395. Authors speak about the abundant presence of livestock in mountain area and the role of prey where anti-predator measures for livestock protection were not yet implemented at the time. Did you think that is just a consequence of mismanagement? Or there is also a lack of knowledge and poor communication from the competent authorities? I think that you must underline this aspect.
This specific topic was extensively analyzed in the two cited papers (Meriggi et al. 2020 and Dondina et al. 2014). Livestock mismanagement, for example the lack of adequate protective measures or calf births on pastures, could have been attributable to farmers inability (both economic and practical) or negligence in addressing the issue. Probably it was not a problem of lack of knowledge, as wolf presence in the area was stable at least since a couple of decades (or more); on the other hand, only in recent years local administrations have started to provide practical support to farmers with protective systems. In contrast, depredation refunds have always been available. Considering the complexity of the issue and that it was already examined in depth in related researches, we preferred to overlook it avoiding oversimplified explanations. The undeniable fact is that, during the first years of the study, livestock was seasonally more easily accessible for wolves if compared to present.
Line 409-412. All these information about the environmental conditions now, with the development of new technologies such as GIS and Remote Sensing, represent an incredible source of data. Not only for research interests but also for the availability of health information (for example about zoonoses in wildlife) and the important impact to Public Health (https://doi.org/10.3390/rs12213542).
Thanks for this interesting paper.
Line 414-423. In this part, I advise you also to do a brief consideration to climate change that represent an important factor impactful on the habitat conditions that reflecting on prey presence and, consequently of wolf presence (https://doi.org/10.1007/s00442-017-4017-y).
Ok, we added the suggested brief consideration about climate change and future research topics (L 459-462).
Conclusion. What will be the future challenges in the scenario you have proposed? The research will be able to answer to new human-livestock-wildlife interface questions?
We added a final consideration on the (plausible) future challenges related to wolf expansion in human-dominated landscape (L 501-503).